# Driver Injury from Vehicle Side Impacts When Automatic Emergency Braking and Active Seat Belts Are Used

**DOI:** 10.3390/s23135821

**Published:** 2023-06-22

**Authors:** Min Li, Daowen Zhang, Qi Liu, Tianshu Zhang

**Affiliations:** 1School of Automobile and Transportation, Xihua University, Chengdu 610039, China; 2Vehicle Measurement Control and Safety Key Laboratory of Sichuan Province, Xihua University, Chengdu 610039, China; 3Engineering, Computer and Mathematical Sciences, The University of Adelaide, Adelaide 5005, Australia

**Keywords:** AEB, intelligent vehicle, driver injury, active seatbelt, side restraint system

## Abstract

As an advanced driver assistance system, automatic emergency braking (AEB) can effectively reduce accidents by using high-precision and high-coverage sensors. In particular, it has a significant advantage in reducing front-end collisions and rear-end accidents. Unfortunately, avoiding side collisions is a challenging problem for AEB. To tackle these challenges, we propose active seat belt pretensioning on driver injury in vehicles equipped with AEB in unavoidable side crashes. Firstly, records of impact cases from China’s National Automobile Accident In-Depth Investigation System were used to investigate a scenario in which a vehicle is impacted by an oncoming car after the vehicle’s AEB system is triggered. The scenario was created using PreScan software. Then, the simulated vehicles in the side impact were devised using a finite element model of the Toyota Yaris and a moving barrier. These were constructed in HyperMesh software along with models of the driver’s side seatbelt, side airbag, and side curtain airbag. Moreover, the models were verified, and driver out-of-position instances and injuries were evaluated in simulations with different AEB intensities up to 0.7 g for three typical side impact angles. Last but not least, the optimal combination of seatbelt pretensioning and the timing thereof for minimizing driver injury at each side impact angle was identified using orthogonal tests; immediate (at 0 ms) pretensioning at 80 N was applied. Our experiments show that our active seatbelt with the above parameters reduced the weighted injury criterion by 5.94%, 22.05%, and 20.37% at impact angles of 90°, 105°, and 120°, respectively, compared to that of a conventional seatbelt. The results of the experiment can be used as a reference to appropriately set the collision parameters of active seat belts for vehicles with AEB.

## 1. Introduction

Traffic collisions cause numerous casualties. Numerous types of collisions arise with complex characteristics and varying deadliness [1]. Side crashes, a common form of traffic collision, are twice as deadly as frontal crashes [2]. Equipping vehicles with autonomous emergency braking (AEB) systems can reduce traffic accidents by 27–38% [3]. However, in the precrash phase, an AEB intervention can cause occupants to move out of position (OOP) within the vehicle despite the presence of a passive seat belt restraint. Moreover, AEB cars have a significantly lower crash avoidance rate in side collisions than in other collisions, and the risk of occupant injury or death is greater [4]. To better protect the occupant during vehicle side impacts, more research is required regarding OOP occupants with passive seat belt restraints in the AEB precrash phase and regarding OOP occupants and driver injury patterns with or without active seat belt restraints in the crash phase.

Various researchers have studied AEB systems. Yang et al. [5] established the upper-layer fuzzy neural network model of AEB-P model based on time to collision and braking safety distance in order to reduce the discomfort caused by emergency braking to passengers and drivers. Salvatore et al. [6] compared rear-end collisions with and without AEB and found that both occupant OOP and whipping injuries were greater if AEB was applied. Hang et al. [7] proposed a real-time automatic emergency system (RTAEB) to improve the sensitivity of AEB system control for rear-end collisions. Based on frontal collisions, Huang et al. [8] collected time-to-collision and conflict distance data to develop a conflict distance model that could distinguish between frontal and side collisions. Cicchino [9] suggested that an AEB system with pedestrian detection could reduce crash risks by 25–27%. Song et al. [10] and Zhou et al. [11] optimized the AEB for different crashes to further improve driver and occupant protection.

Krouse et al. [12] and Wu et al. [13] researched active seat belts and demonstrated that they have a significant protective effect after a vehicle has taken steering or avoidance measures. Song et al. [14] verified the control ability of active seat belts through bench tests. Good et al. [15] and Ito et al. [16] studied the effect of reversible pretensioning seat belt parameters on the correction of OOP occupant posture, and the results revealed that a greater seat belt pretensioning force was more effective than that with an ordinary seatbelt for correcting OOP. Sun et al. [17] found that active seat belts could improve system safety without greatly affecting comfort.

In studies of combined active seat belts and AEB systems, Wang [18] established an integrated restraint system in overtaking emergency conditions and used weighted injury criterion (WIC) values to demonstrate that damage to some parts of a dummy was reduced by the integrated action of the AEB system and the active pretensioner. Wang et al. [19] analyzed the effect of AEB triggering and braking force on occupant posture and injury in rear-end collisions; optimizing the active seat belt parameters could reduce a driver’s neck injuries. Li et al. [20] found that active restraint reduced OOP instances in various seating positions and effectively reduced WIC values in AEB-triggered frontal crash conditions.

In summary, active seat belts can reduce OOP instances during precrash AEB. Conventional driver restraint systems were developed for the normal sitting position in frontal and rear-end conditions [21]. During side impacts, the driver’s upper body tilt increases OOP instances and the complexity of the driver’s dynamic response, increasing the risk of driver injury [22,23]. Few scholars have explored the protective effect of active seat belts for different vehicle–vehicle side impact angles with AEB. Therefore, it is important to analyze the injury to OOP drivers in unavoidable side crashes, in vehicles equipped with AEB, specifically in relation to active seat belts, to improve driver protection. The contributions of this paper are as follows.

In this paper, we designed a side impact scenario. Specifically, the model used real accident cases in a Chinese database, the National Automobile Accident In-Depth Investigation System (NAIS), and we determined the collision location and speed. This allowed us to establish a scenario where the main vehicle triggers the AEB when it is hit by a side-approaching vehicle.We constructed a finite element model of a whole vehicle. Based on National Highway Traffic Safety Administration (NHTSA) report V06583 [24] regarding a Toyota Yaris side impact, we used finite element software to establish and validate the whole vehicle finite element model.We designed a driver’s side airbag, side curtain airbag, and seat belt restraint system, and conducted a series of model validations. The design of these components provided strong support for the subsequent side impact experiments.The effects of passive and active seat belts on the driver OOP instances and injury were realized for different braking decelerations and side impact angles. Among other things, the seatbelt pretightening force and moment were determined using orthogonal tests. The results can be helpful for optimizing the design of active seat belts. Additionally, these findings provide theoretical support for methods of increasing the protection of drivers and passengers under different side impact angles.

## 2. Materials and Methods

Side crashes when AEB collision avoidance fails are complex situations in which active and passive systems act together. This study investigated the preparation of active seat belts in protecting the driver in a side impact in five steps, including data sources of the accident, the whole vehicle finite element model for side impacts, a side restraint system, the principles of active seat belts, and driver injury assessment. These components collectively aim to provide a comprehensive understanding of side collisions and the potential measures for reducing driver injuries.

### 2.1. Data Sources

Established in 2011, NAIS is an in-depth traffic accident investigation system managed by China’s State Administration of Market Supervision and Administration in partnership with numerous universities, research institutions, and traffic accident forensic centers. The database contains reports on vehicle collisions (particularly those with fatalities or caused by vehicle safety defects) and fires (particularly those involving electric or hydrogen vehicles); the data collection standards were developed in reference to the in-depth accident investigation systems established by the United States (NHTSA) and Germany (Bundesanstalt für Straßenwesen). Data on incidents in different regions and geographical environments are stored at eight locations in China, and various vehicle types and crash patterns are included in the records. Hence, the dataset contains a representative sample of traffic accidents in China [11].

In this paper, 169 cases of vehicle–vehicle side impact collisions were selected from NAIS with the following criteria: occurrence at an unsignalized intersection, significant vehicle deformation, only two parties involved in the collision, and both vehicles being passenger cars. A driver is more likely to be seriously injured if the proximal end (left side) is hit than if the distal end (right side) is hit. With reference to the relevant literature [22,25], a precrash scenario was established using PreScan (TNO, Eindhoven, Netherlands) software as follows: the impacted car is driving through an intersection at a constant speed of 34 km/h when a car in front (initially driving at the same speed) rapidly brakes to a stop to avoid a pedestrian; the impacted car’s AEB system activates, and a car approaching the intersection from the right at 55 km/h impacts the impacted car on its left side at the A-pillar position. The constructed collision scenario is shown in Figure 1.

### 2.2. Side Impact Whole Vehicle Finite Element Model

A finite element model of the Toyota Yaris sedan was developed by the National Crash Analysis Center at George Washington University by using reverse engineering. The model includes an engine, chassis, suspension, steering, body, interior, and seat components. This model was adopted for the simulations in this study, and the model’s composition is presented in Table 1. The main parameters of the computer-aided engineering (CAE) model and the actual vehicle are presented in Table 2; the shape, parts, and material properties of the model and the actual vehicle are consistent.

A model of a mobile barrier dolly was used for the simulated side impacts. The finite element model of the mobile barrier was produced by Livermore Software Technology Corporation (LSTC) and is freely available to download on the LSTC website. The model was designed to enable the obtainment of accurate results with minimal calculation time. The front of the main body frame is an aluminum honeycomb structure with a width of 1252 mm. Including this aluminum front, the total length of the mobile barrier is 4115 mm. The axle base is 2592 mm, and the total mass is 1361.5 kg. A front view of the model including its dimensions and height above the ground is displayed in Figure 2.

The front projection of the mobile barrier simulates the front bumper projection of an actual car. The height of the front bonnet edge is also similar to that of an actual car. The simulated barrier is displayed in Figure 3.

### 2.3. Side Restraint System

Primer software (Ansys, PA, USA) is used for crash safety occupant restraint systems to build finite element restraint system models, which are widely used for occupant protection studies in vehicles. This includes driver positioning, seat belt establishment, airbag folding, seat compression, etc.

The dummy and seat model

To improve the accuracy of simulation results, the posture of the dummy is adjusted according to the positioning standards of the dummy in the real vehicle test report (shown in Figure 4a). Due to the gravitational effect of the occupant, the seat will undergo deformation. Therefore, preloading is applied to the driver seat in the simulation model to simulate the deformation that occurs when the driver contacts the seat. The preloaded seat is shown in Figure 4b.

2.Establishment of the seatbelt models

First, accurate mannequins and pressurized seats were introduced into the Primer software program, and the seatbelt creation method was defined. Second, the retractable slide of the seatbelt between the reel and slip ring was simulated using Unit 1D, and the part of the seatbelt that contacts the dummy’s chest and hip (shoulder strap, belt) was simulated using Unit 2D. Finally, the “side harness module” with a contact type of “surface contact,” was imported to the HyperMesh finite element preprocessor to set the remaining parameters.

3.Establishment of the side airbag and side air curtain

A side airbag mainly protects the driver’s chest and pelvis, whereas a side curtain airbag mainly protects the driver’s head. The parameters of the side airbag were obtained from the parts manufacturer; with these parameters, the profile of the side airbag (shown in Figure 5a) was drawn in the CATIA (Dassault Systèmes, Vélizy-Villacoublay, France) modeling software and then imported into HyperMesh (Altair Engineering, Troy, MI, USA). The two planes were then flattened and stitched together to produce vents. The results of the simulated and actual airbag static point explosion tests were consistent, validating the airbag model. The side curtain airbag was modeled similarly to the side airbag. The roll fold mode (Figure 5b) was adopted because it has a favorable airbag orientation and the direction of expansion can be easily controlled. In the actual test, the electronic control unit of the curtain airbag makes a judgment and activates the gas generator at between 6 and 13 ms, and the airbag then fills through a small hole in the gas diffusion pipe at between 20 and 30 ms. In the simulation, the gas generator is activated and the air curtain begins to inflate by *t* = 10 ms, the curtain airbag is halfway full by *t* = 15 ms, and the curtain airbag is fully inflated by *t* = 30 ms. Hence, the test and simulation results are consistent, validating the model.

4.Validation of the model by load constraints

To verify the model for various collision conditions, the model’s performance was investigated for various load constraints. Figure 6 presents the energy changes during a simulated collision; the kinetic energy, internal energy, eroded hourglass energy, and total energy curves are smooth as the system kinetic energy is converted into endogenous energy. No nonphysical sudden increases in hourglass energy or mass occur. The hourglass energy loss is less than 5% of the total energy, and the initial mass increase is less than 10 kg. Therefore, the finite element model can be used in subsequent experiments. (Since the ratio of Total Energy/Initial Energy always remains close to 1, the green line differs significantly from other energy curves and may not be easily discernible).

### 2.4. Active Seatbelt Principles

When an active seatbelt begins pretensioning, a DC motor starts and a clutch mechanism engages the reel core shaft, which rotates to retrieve the seatbelt’s webbing, tightening the seatbelt until the motor is overloaded, and then stops [26]. Passengers lean forward during braking deceleration. When the seatbelt’s pretensioning sensor recognizes the braking signal, the seatbelt executes tightening, confining the passenger to the seat and eliminating any gap between the seatbelt and the passenger.

An active seatbelt retractor for pretensioning during braking was modeled for the aforementioned model seatbelt (Figure 7). In the figure, direction (1) represents the movement of the webbing when the retractor is pretightened. The retractor pulls the webbing by rotating the reel, and the seatbelt then translates in direction (2), tightening the shoulder belt and constraining the occupant during braking. To reduce computational complexity, the initial loading moment of the braking deceleration was set as the beginning of the simulation (*t* = 0 ms), and the total pretensioning time was 40 ms.

### 2.5. Driver Injury Assessment Indicators

One of the common injuries in traffic accidents is head injury [27]. Head injury severity is often determined on the basis of the head tolerance curve and the head injury intensity index. Head injury severity is quantified using the head injury criterion (*HIC*), which is calculated by integrating head acceleration over a time period of up to 36 ms. The risk of a dummy head injury at abbreviated injury scale level 3 or greater (*AIS*3+) is as follows:(1)$$P_{head}\left(AIS3+\right)=\emptyset \left(\frac{\ln\left({HIC}_{36}-7.45231\right)}{0.73998}\right)$$

The *HIC* is calculated as follows:(2)$$HIC={\max\left[\frac{1}{t_{2}-t_{1}}\int_{t_{1}}^{t_{2}}a\left(t\right)d_{t}\right]}^{2.5}\left(t_{2}-t_{1}\right)$$

In the formula, n = 2.5 is a weighting index, and *t*_1_ and *t*_2_ represent two times on the head acceleration curve, *a*(*t*).

Side impact chest injury evaluation indexes include the thoracic trauma index, chest compression quantity, and the viscous criterion (*VC*). In this paper, the *VC* was used to evaluate chest injuries and was calculated as follows:(3)$$VC=V\times C=\frac{d\left[D\left(t\right)\right]}{dt}\times \frac{D\left(t\right)}{D}$$
where *V* represents the deformation speed and *C* represents the chest compression. *VC* values are in m/s. Fatal chest injuries may occur if the *VC* is greater than or equal to 1.0 m/s.

Abdominal injury was evaluated using the abdomen peak force (*APF*) index at a threshold of 2500 N. The risk of an abdominal injury at *AIS*3+ can be calculated from the abdominal resultant force as follows:(4)$$P_{abdomen}\left(AIS3+\right)=\frac{1}{1+e^{6.04044-0.002133\times abdominal\;force}}$$

In this study, the risk of pelvic injury was assessed on the basis of the pubic force. Similarly to the method for abdominal forces, the probability of an *AIS3*+ pelvic injury can be calculated as follows:(5)$$P_{pelvis}\left(AIS3+\right)=\frac{1}{1+e^{7.5969-0.0011*Pubic\;force}}$$

The chest compression threshold was adjusted to 44 mm in accordance with FMVSS regulations. The overall *WIC* for dummy damage is thus calculated as follows:(6)$$WIC=0.3\left(\frac{RDC}{44}+\frac{VC}{1.0}\right)+0.2\left(\frac{APF}{2.5}+\frac{PSPF}{6}\right)$$
where *VC* is the viscous criterion (m/s), *RDC* is the rib compression of the chest (mm), *APF* is abdomen peak force (kN), *PSPF* is pelvis or pubic force (kN), and 0.3 and 0.2 are weight values.

## 3. Results

Experiments were conducted using the developed models to compare the effect of braking strength on driver OOP and injury for both conventional and active seatbelts and at various angles of contact.

### 3.1. Model Validation

The vehicle testing simulations were performed in accordance with US Federal Motor Vehicle Safety (FMVSS) regulation 214 (FMVSS214) [28]. The regulation states that the main vehicle should be stationary, and the mobile barrier should crash into the vehicle from the left side at 61.1–62.7 km/h. In the simulation, the speed of the barrier was set to 61.8 km/h, and its impact direction was 27° to the central axis of the car. During tests, dummy were placed on both the driver’s seat and in the back row on the driver’s side. The initial collision position corresponds to that in a real car test. The mobile barrier and Yaris finite element model were placed in the same plane, and the left edge of the mobile barrier was aligned with the left edge of the Yaris-based finite element model, as shown in Figure 8.

Figure 9 presents the deformation of each component on the left side of the vehicle. The deformed parts include the left front wing plate, left front and rear door, threshold beam, B pillar, and window glass. The deformed parts and their corresponding deformation depths of the finite element model and the actual vehicle are largely consistent [29]. For the mobile barrier, most deformation occurred at the front aluminum honeycomb, and the simulated and actual amounts of deformation were consistent.

Figure 10 reveals that the simulated acceleration values at the bottom and center of pillar B have similar trends, peaks, and troughs to those of the actual test. The times of these peaks and troughs differ slightly because the crushing deformation of the door and the B pillar caused a fracture failure during the actual collision but not the simulation. According to Figure 10a, the maximum peak or trough of difference between the simulated and actual acceleration results at the bottom of pillar B is approximately 13.93 g, with an error range of about 10.53%. In Figure 10b, the maximum peak or trough of difference between the simulated and actual acceleration results at the center of pillar B is approximately 18.20 g, with an error range of about 14.75%. The errors are all within 15%, so the simulated and actual results are consistent overall.

### 3.2. Driver Displacement Analysis

The speed of the impacted vehicle in the side impact scenario is 34 km/h (less than 48 km/h). To mimic actual road conditions, the maximum sliding friction coefficient was set to 0.7 (equivalent to a maximum braking speed of 0.7 g). To represent AEB attempting to avoid a collision or reduce the severity of the accident, 0.6 g single-stage braking, 0.7 g step braking, or 0.4 g + 0.6 g graded braking was applied. The displacement of vehicles for a given braking time was determined through experiments. To ensure that the point of collision of the two vehicles was the same in all simulations, the initial positions of the vehicles were modified in accordance with the braking strength. Simulations revealed that driver displacement during braking is mainly in the *x* direction; displacement in the *y* and *z* directions was small. Therefore, *y* and *z* displacement was ignored.

#### 3.2.1. Pre-Collision Restraint by the Conventional Seatbelt

An ordinary seatbelt applies no pretensioning force in the braking stage. During braking deceleration, the dummy moves forward due to inertia. Because the head is not directly restrained by the seatbelt, its movement is greater than that of the body. In addition, the head is also subjected to tension and torque from the cervical spine while leaning forward. Hence, the head tends to rotate and slightly nod as the torso tilts forward. The lower limbs of the dummy also have inertia but are restrained from moving forward by the seatbelt; only small movements of the hips, thighs, and calves occur.

Table 3 compares the head centroid displacement, lateral displacement of the first thoracic vertebra (T1), RDC, APF, and PSPF for different braking strengths. Each index is positively correlated with braking strength, but VC is not necessarily positively related to the maximum chest compression, which is affected by the chest deformation rate. Compared with single braking, staged braking causes less forward tilting and chest compression for the same braking time. Calculations revealed that the head HIC was 46.88 at 0.4 *g* + 0.6 *g* braking, 42.7 at 0.6 *g* braking, and 42.3 at 0.7 *g* braking.

#### 3.2.2. Active Seatbelt Restraint during a Collision

Figure 11 presents the partial movement of the driver’s posture under the action of active seatbelt pretensioning. The simulation reveals that pretensioning reduces dummy dislocation during braking. During impact, the left torso of the dummy shifts from its point of contact with the door trim panel, increasing its distance from the maximum deformation position of the door trim panel and reducing damage to the dummy. In addition, the backward movement of the contact location on the left side of the dummy increases the contact area between the left side of the dummy and the side airbag, improving the protection effect of the side airbag.

### 3.3. Driver Injury at Different Side Impact Angles

To study the effect of the angle on the side impact results, 90°, 105°, and 120° collisions were simulated. The contact location was constant for all collision angles. Changing the side impact angle of the moving barrier affected both the deformation volume and the maximum deformation area of the vehicle.

Orthogonal experiments to optimize the WIC were designed for the following variables: active seatbelt pretensioning force, pretensioning time, and collision angle. The SPSSAU software program (version 23.0, Beijing Qingsi Technology, Beijing, China) was used to perform a range analysis of the data in Table 4 to obtain the optimal parameters and revealed that 80 N pretensioning, a pretensioning time of 0 ms, and an 105° impact angle was the optimal combination. Therefore, in subsequent simulations of side collisions, 80 N of pretensioning was applied at 0 ms.

#### 3.3.1. Comparison of Driver Damage in 90° Collisions with and without Active Seatbelts

RDC (Figure 12a), APF (Figure 12b), and PSPF (Figure 12c) were compared for the conventional seatbelt and pretensioned active seatbelt in a 90° collision. The pretensioned active seatbelt had lower RDC, APF, and PSPF curves and earlier peak injury times than did the conventional seatbelt. With the conventional seatbelt, the maximum and minimum chest compression was 34.69 and 32.33 mm, respectively; the maximum and minimum VC was 0.492 and −0.426 m/s, respectively; the APF was 638.87 N; and the PSPF was 3908.04 N. According to Formula (6), the WIC was 0.565. With the pretensioned active seatbelt, the maximum and minimum chest compression was 32.41 and 30.99 mm, respectively; the maximum and minimum VC was 0.475 m/s and −0.412 m/s, respectively; the APF was 604.19 N; and the PSPF was 3737.84 N. The WIC was 0.536. Hence, the pretensioned active seatbelt reduced chest rib compression by 6.57% and the WIC by 5.13%.

#### 3.3.2. Comparison of Driver Damage in 105° Collisions with and without Active Seatbelts

The RDC (Figure 13a), APF (Figure 13b), and PSPF (Figure 13c) were compared for the conventional seatbelt and pretensioned active seatbelt in a 90° collision. The pretensioned active seatbelt had lower RDC, APF, and PSPF curves and earlier peak injury times than did the conventional seatbelt. With the conventional seatbelt, the maximum and minimum chest compression values were 35.95 and 33.34 mm, respectively; the maximum and minimum VC was 0.566 and −0.462 m/s, respectively. The APF was 914.76 N, and the PSPF was 2900.96 N. The WIC was 0.585. With the pretensioned active seatbelt, the maximum and minimum chest compression values were 29.62 and 27.86 mm, respectively; the maximum and minimum VC values were 0.317 and −0.357 m/s, respectively. The APF was 884.63 N, and the PSPF was 2745.01 N. The WIC was 0.459. Using the pretensioned active seatbelt reduced chest rib compression by 17.61% and the WIC by 21.53%.

#### 3.3.3. Comparison of Driver Damage in 120° Collisions with and without Active Seatbelts

The RDC (Figure 14a), APF (Figure 14b), and PSPF (Figure 14c) were compared for the conventional seatbelt and pretensioned active seatbelt in a 90° collision. The pretensioned active seatbelt had lower RDC, APF, and PSPF curves and earlier peak injury times than did the conventional seatbelt. With the conventional seatbelt, the maximum and minimum chest compression values were 37.51 and 33.84 mm; the maximum and minimum VC values were 0.607 and −0.464 m/s, respectively; the APF was 1251.97 N; and the PSPF was 2217.74 N. The WIC was 0.608. With the pretensioned active seatbelt, the maximum and minimum chest compression values were 32.61 and 27.77 mm, respectively; the maximum and minimum VC values were 0.368 and −0.342 m/s, respectively; the APF was 1207.25 N; and the PSPF was 1977.77 N. The WIC was 0.496. The pretensioned active seatbelt reduced chest rib compression by 13.06% and the WIC by 18.42%.

## 4. Discussion

### 4.1. Validation of the Results

Xu et al. [30] performed real vehicle tests on occupant displacement and injury during braking with two volunteer occupants and a seatbelt retractor with active control. Active control of the retractor significantly reduced head, T1, and shoulder displacement in the volunteers. This result is consistent with the simulation results in Section 3.2.2. Luo et al. [31] conducted simulated frontal collision experiments with Hybrid III dummies and reported that certain pretensioning in the precollision could control the torso movement and chest compression of the dummy; this result is also consistent with the simulation results in this paper. Luo et al. [31] also reported that optimizing the preload time and force can further increase occupant protection. In this study, the combination of 80 N of pretensioning and a time of 0 ms was identified as optimal for an orthogonal test (Section 3.3) of dummy injuries at three typical dangerous side impact angles. The results indicate that the active seatbelt has a strong protective effect.

### 4.2. Advantages of Research

Arathanaikotti et al. [32] inferred that side impact has a greater impact on occupant injury severity, and safety devices in the vehicle such as traditional seat belts may not effectively mitigate injuries from side impacts. Imler et al. [33] investigated the effects of relative velocity and restraint systems on injuries sustained by occupants in vehicle-to-vehicle side impacts, and demonstrated that belted occupants experienced lower levels of injury compared to unbelted occupants. David et al. [34] compared and analyzed four combinations of vehicles with and without side airbags and side curtain airbags. They found that the side curtain airbags alone reduced head acceleration by 42%, the side airbags alone reduced chest compression by 54%, and using both further reduced occupant injury values. The above studies provided a reference aiming to optimize the parameters of active seatbelts, side airbags, and side curtain airbags. Therefore, we focus on the research of active seat belts for driver protection in the event of a side impact, especially regarding driver injuries that occur when the striking vehicle experiences a change in impact angle.

In Section 3.3, injuries in collisions with active seatbelts and conventional seatbelts at different side impact angles were evaluated, revealing that an active seatbelt can significantly reduce dummy injuries. For the pretensioned seatbelt, the PSPF for the 120° collision was 1760.07 N less than that for the 90° collision; the corresponding reduction for the conventional seatbelt was 1690.3 N (69.77 N smaller). Furthermore, the APF was 10.75 N lower with the active seatbelt than that with the conventional seatbelt. On the basis of the RDC, the risks of a chest injury greater than AIS3+ for the 90°, 105°, and 120° angles are 9.96%, 11.05%, and 12.54%, respectively, when a conventional seatbelt is worn.

Figure 15 presents the RDC for various collision angles in a conventional seatbelt condition. At 105°, the deformation of the vehicle is mainly concentrated in the left front wheel hub, tire and left front fender, and the interior door plate collides with the upper arm, compressing the upper ribs. In a 120° collision, the front door deforms faster, increasing the peak chest compression. The collision between the lower arm and the door greatly compresses the lower ribs and subsequently compresses the chest. The analysis in Section 3.2.2 reveals that the active seatbelt reduces this compression. This seatbelt had the best protective effect for a 105° impact angle; chest compression was reduced by 17.61%.

### 4.3. Evaluation of Model and Method

#### 4.3.1. Pros and Cons of Model

In this paper, there are two pros of the proposed finite element model. (1) We used the coupled approach of Prescan-HyperMesh-LS-DYNA software to build a vehicle collision model that integrates active and passive safety. Firstly, the braking scenario was built with the help of Prescan. The AEB provided during this stage could reduce the risk of misoperation to the greatest extent and improve the veracity of deceleration. Then, we constructed the corresponding finite element model for a whole vehicle side impact in HyperMesh. Finally, using the LS-DYNA software, we simulated and calculated the movement of the vehicle, the movement of the driver, and the changes in their injuries based on the deceleration provided by the AEB. (2) The active seatbelt we designed not only considered the protection effect on the driver but also explored the sensitivity ranking of key parameters through orthogonal experiments. We identified the optimal experimental group for the best parameters, which can improve driving comfort.

However, the model we established also has some drawbacks. The active muscle force of the human body was not simulated in the model. The driver side restraint system model constructed in this paper only used the ES-2re dummy model. Therefore, there are differences in the motion response of the dummy compared to that of a real driver during braking. In future research, active human body models such as the OpenSim model can be considered for simulation.

#### 4.3.2. Insufficient of Method

The active seatbelt parameters were optimized only for three side impact angles. Other factors affecting the side impact and injuries, such as impact speed, driver age, and driver gender, were not considered but could be investigated in future studies.Moreover, the orthogonal experiment design can only be used for a fixed range of values; a future study could use an optimization algorithm to identify the optimal parameters.

## 5. Conclusions

This study mainly investigated differences in driver injury in collisions with different AEB strengths and side impact angles. With the goal of combining active and passive safety mechanisms, an active seatbelt model was established to explore the influence of active seatbelts on driver injury during an AEB intervention for dangerous side impact angles. The results are summarized as follows:

(1) When the vehicle triggers AEB, driver displacement occurs. The amount of dislocation was positively correlated with the AEB deceleration magnitude, and head displacement was greater than thoracic vertebra displacement. Driver injury was positively correlated with braking deceleration. Active seatbelts can reduce driver dislocation more effectively than ordinary seatbelts can.

(2) Active seatbelts can effectively reduce driver injury compared with conventional seatbelts. In 90° collisions, active seatbelts reduced the RDC by 6.57% and the WIC by 5.13%. In 105° collisions, active seatbelts reduced the RDC by 17.61% and the WIC by 21.53%. In 120° collisions, active seatbelts reduced the RDC by 13.06% and the WIC by 18.42%.

(3) The effect of the active seatbelt on driver injury differed for various side impact angles. In a 120° collision, the active seatbelt decreased the RDC and WIC more than in the 105° and 90° collisions did. However, for these impact angles, the active seatbelt can improve protection from the side airbag and can also cause the driver’s left contact position to move from the maximum deformation position of the door, reducing driver injury.

## Figures and Tables

**Figure 1 sensors-23-05821-f001:**
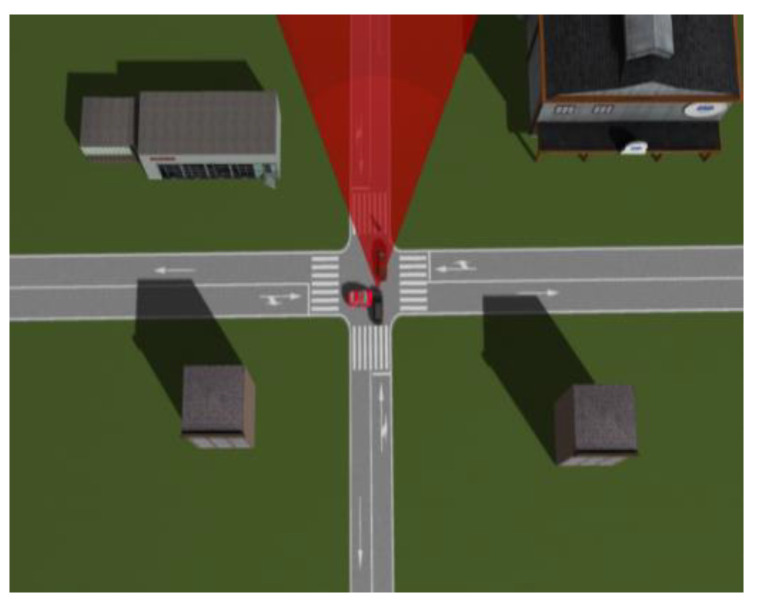
Side collision scenario description.

**Figure 2 sensors-23-05821-f002:**
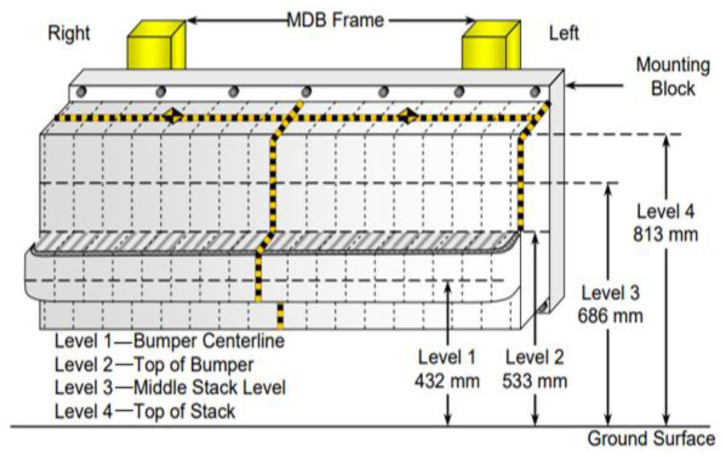
Front structural parameters of a moving barrier.

**Figure 3 sensors-23-05821-f003:**
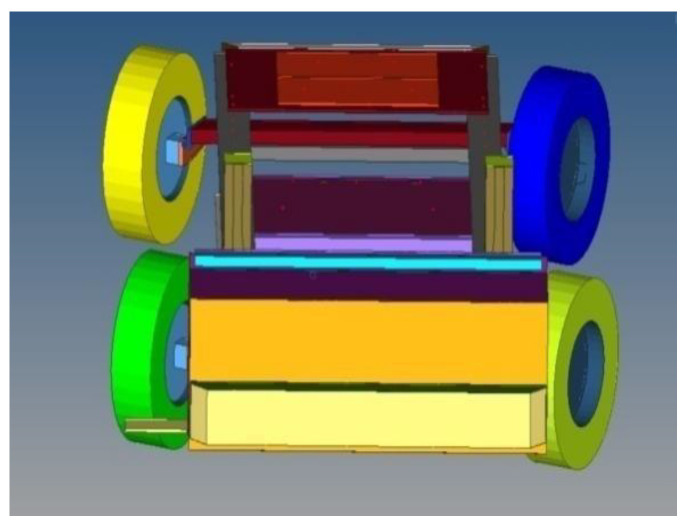
Simulated mobile barriers.

**Figure 4 sensors-23-05821-f004:**
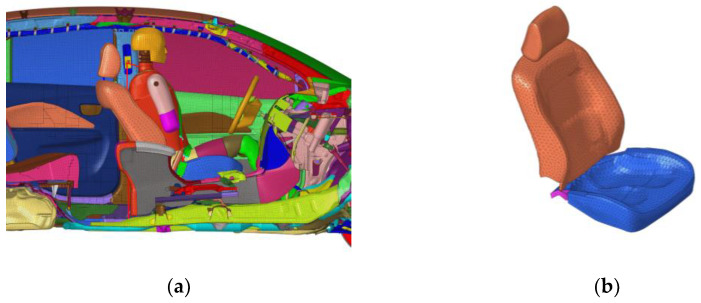
(**a**) Adjusted dummy posture model; (**b**) preloaded seat.

**Figure 5 sensors-23-05821-f005:**
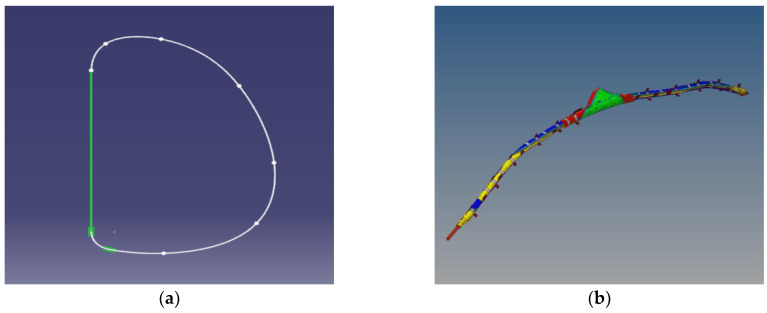
(**a**) Side airbag contour; (**b**) side air curtain folding mode.

**Figure 6 sensors-23-05821-f006:**
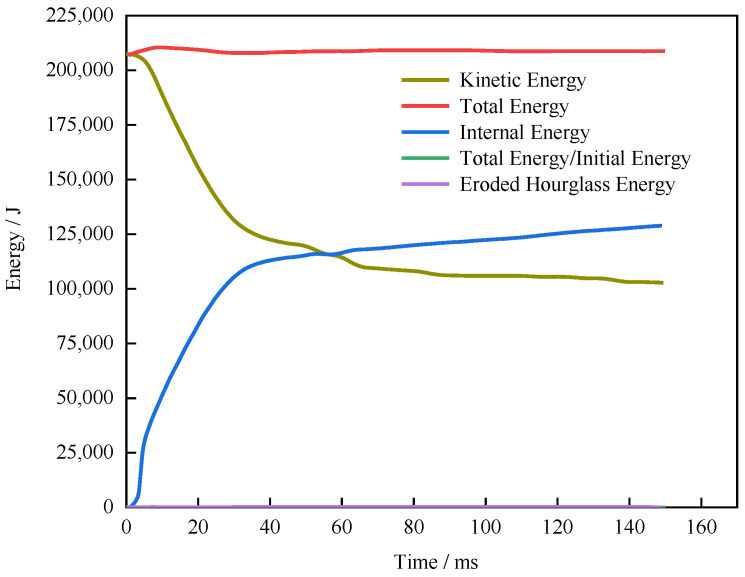
Energy changes in the side impact finite element model.

**Figure 7 sensors-23-05821-f007:**
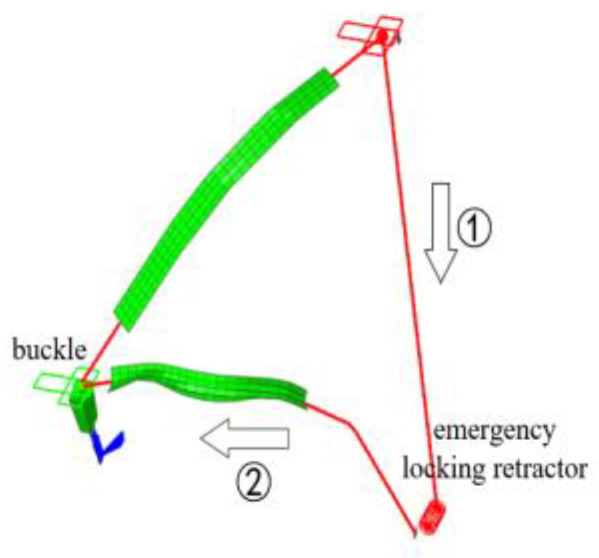
Finite element model of the active seatbelt.

**Figure 8 sensors-23-05821-f008:**
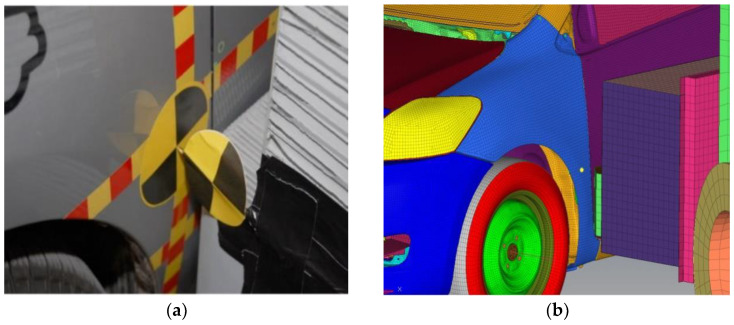
Initial collision positions in actual tests and in the simulation. (**a**) Actual car crash benchmark setup; (**b**) finite element model collision alignment.

**Figure 9 sensors-23-05821-f009:**
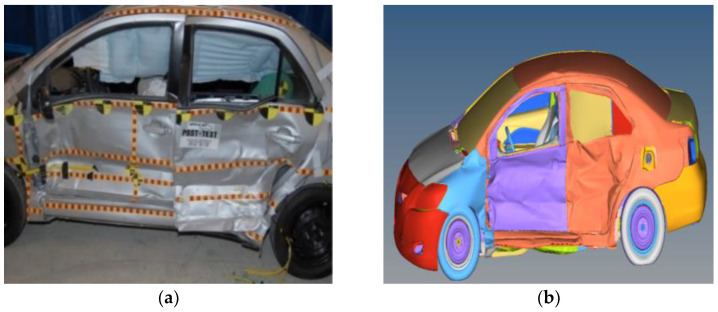
Collision deformation for the actual vehicle and the finite element model. (**a**) Actual vehicle; (**b**) finite element model.

**Figure 10 sensors-23-05821-f010:**
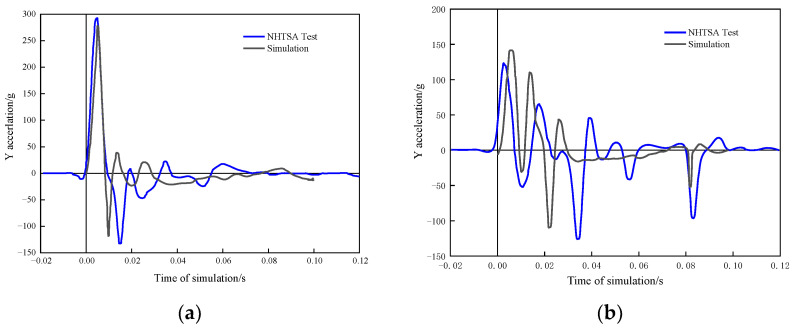
(**a**) Simulated and actual acceleration values at the bottom of the B pillar; (**b**) simulated and actual acceleration values at the middle of the B pillar.

**Figure 11 sensors-23-05821-f011:**
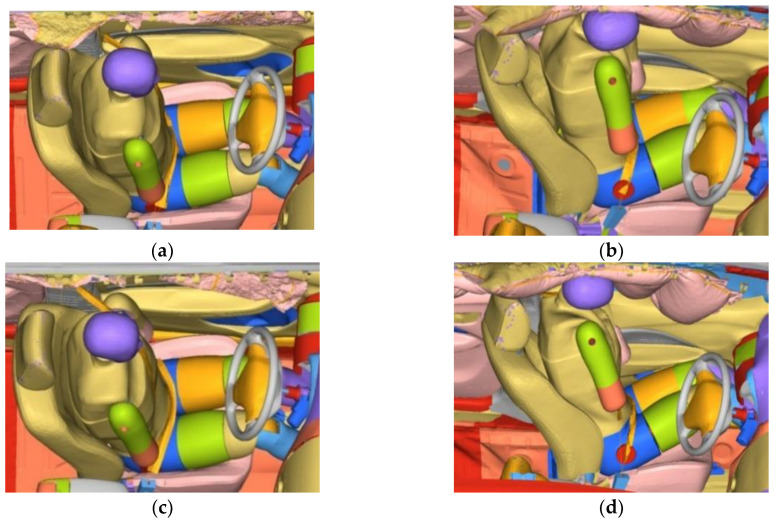
Driver posture during the collision for the conventional and pretensioned active seatbelts. (**a**) Conventional seatbelt at 80 ms; (**b**) conventional seatbelt at 160 ms; (**c**) pretensioned active seatbelt at 80 ms; (**d**) pretensioned active seatbelt at 160 ms.

**Figure 12 sensors-23-05821-f012:**
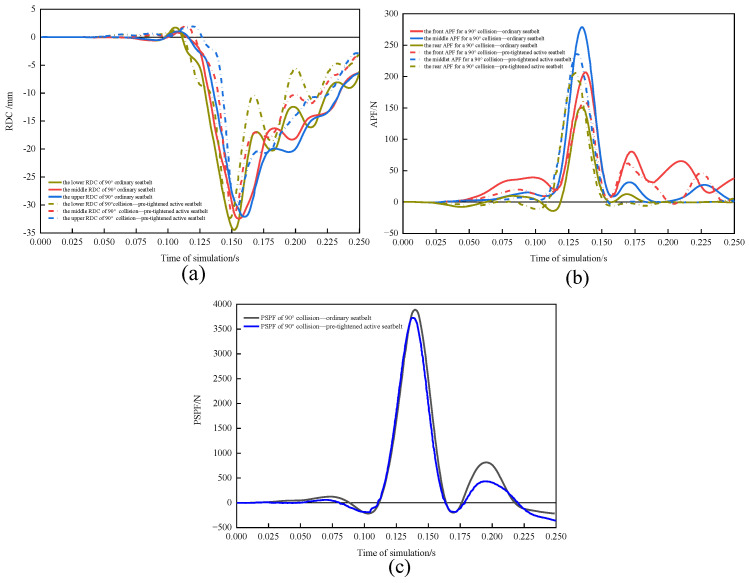
(**a**) RDC, (**b**) APF, and (**c**) PSPF for a 90° collision for the two seatbelts.

**Figure 13 sensors-23-05821-f013:**
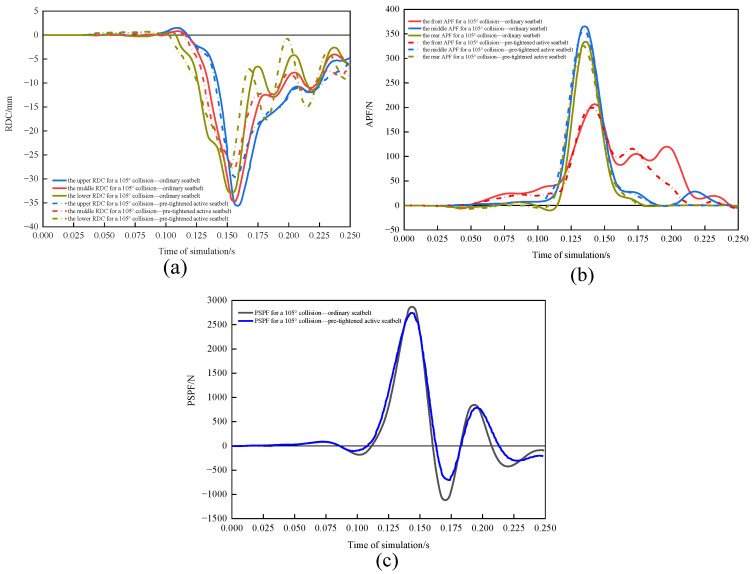
(**a**) RDC, (**b**) APF, and (**c**) PSPF for a 105° collision for the two seatbelts.

**Figure 14 sensors-23-05821-f014:**
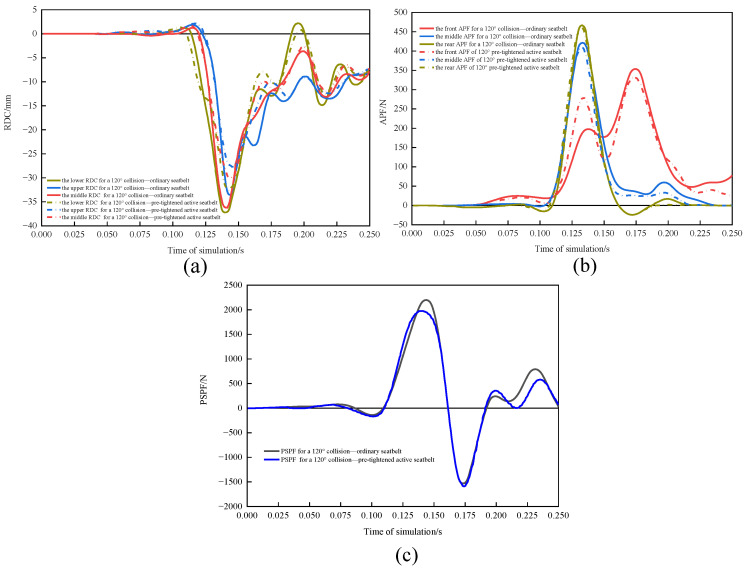
(**a**) RDC, (**b**) APF, and (**c**) PSPF for a 120° collision for the two seatbelts.

**Figure 15 sensors-23-05821-f015:**
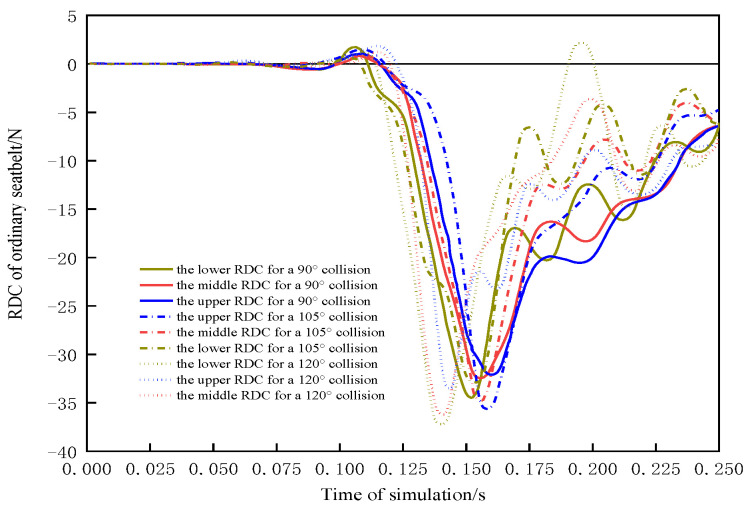
RDC compression for the conventional seatbelt.

**Table 1 sensors-23-05821-t001:** Composition of the Yaris-based finite element model.

Unit	Number	Connections	Number
Node units	1,480,422	Beam unit connections	4425
Shell units	1,250,424	Nodal rigid connections	727
Beam unit	4738	Nodal set connection	20
Entity units	258,887	Rigid body connections	2
-	-	Welded joint connections	4107
-	-	Hinged connections	39

**Table 2 sensors-23-05821-t002:** Parameters of the CAE model and actual vehicle.

Parameters	CAE	Actual Vehicle
Overall vehicle mass (kg)	1100	1078
Pitch inertia of rotation (kg·m^2^)	1566	1498
Transverse inertia (kg·m^2^)	1739	1647
Sway inertia (kg·m^2^)	395	388
Vehicle center of mass X scale (mm)	1004	1022
Vehicle center of mass Y scale (mm)	−4.4	−8.3
Vehicle center of mass Z scale (mm)	569	558

**Table 3 sensors-23-05821-t003:** Driver displacement for different braking intensities.

Brake Strength	Head Centroid Displacement /mm	T1 /mm	Maximum Chest Compression (mm)	Minimum Chest Compression (mm)	Maximum VC (m/s)	Minimum VC (m/s)	APF (N)	Maximum PSPF (N)
0.4 g + 0.6 g	68.747	54.778	26.53	25.05	0.246	−0.206	520.10	838.462
0.6 g	77.576	59.383	29.99	27.07	0.354	−0.241	703.24	1010.15
0.7 g	86.611	68.803	34.69	32.33	0.485	−0.353	865.13	1119.77

**Table 4 sensors-23-05821-t004:** $L_{9}\left(3^{3}\right)$ orthogonal table.

Trial Number	Pretensioning Force (N)	Pretensioning Time (ms)	Collision Angle	T1 Displacement	Trial Number	Pretensioning Force (N)	Pretensioning Time (ms)
1	60	0	90°	49.258	71.362	32.858	0.563
2	80	20	90°	49.217	68.604	33.552	0.581
3	100	40	90°	49.529	71.747	32.593	0.574
4	100	20	105°	36.833	66.632	32.836	0.505
5	80	0	105°	40.826	67.904	29.621	0.459
6	60	40	105°	55.392	73.172	29.894	0.488
7	60	20	120°	50.946	70.6	32.611	0.511
8	80	40	120°	45.073	70.369	30.209	0.499
9	100	0	120°	38.227	66.533	30.670	0.526

## Data Availability

Not applicable.

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
