# Peer review of "Driver Injury from Vehicle Side Impacts When Automatic Emergency Braking and Active Seat Belts Are Used"

_sensors, 2023, doi:10.3390/s23135821_

Round 1

Reviewer 1 Report

The increasing intensity of traffic demands continuous improvements in vehicles, with an increasing focus on safety. Passive vehicle safety remains an important issue. Studies show that the risk of accidents does not decrease with the development of vehicles and the increase in speed. Although modern numerical models, computer-aided design techniques, and the availability of modern materials allow rapid solutions, the fact that they exist shows the complexity of the safety problem. This article develops mathematical and experimental methods for the assessment of driver injuries in a side impact collision with automatic emergency braking and active seat belts. The conclusions discuss the results of the theoretical and experimental studies. The article is very interesting from a side impact crashworthiness point of view and adds to the lack of research on the use of the effects of active seat belt pretension on driver injury in vehicles equipped with automatic emergency braking. The article is well written, and the topic could be of interest to the readers of the journal. The choice of research methodology is appropriate and logical. The keywords are appropriate and the quality of the figures and tables is good.

The paper addresses an important and interesting issue, but there are several places where the paper should be revised and improved.

1) In the case of a side impact, the contact between the dummy and the seat has a major influence. Perhaps a more detailed description of the dummy and the seat could be provided.

 2) Has the impact of head restraints on the injury criteria been assessed?

Overall, my opinion is positive and I would suggest that the authors continue the work.

Minor editing of English language required.

Reviewer 2 Report

The authors investigated differences in driver injury in collisions with different AEB strengths and side impact angles. I have some suggestions on this paper.

1- Please check the sentences, such as ”In the following text, the 91 accident data sources are introduced, the whole-vehicle finite element model for side 92 impacts is validated, a side restraint system is established, the principles of active seat 93 belts are introduced, and driver injuries are evaluated.” It is too rigid.

2- Figure 6: “However, this difference was small, and the simulated and actual results are consistent overall.” Please provide the value of error.

3- Figure 7: This figure is not clear, please enlarge it.

4- Table 3: 0.4+0.6g? It should be 0.4g+0.6g.

English expression is not so good, please check the entire manuscript carefully.

Reviewer 3 Report

In this paper, the authors propose an active seatbelt model to explore the influence of active seatbelts on driver injury during AEB intervention for dangerous side impact angles.

It was a pleasure reviewing this work and I can recommend it for publication after a minor revision. I respectfully refer the authors to my comments below.

1- The abstract must be a concise yet comprehensive reflection of what is in your paper. Please modify the abstract according to the motivation, description, results, and conclusion parts.

2- In the introduction part, show what the originality of your work is.

3- Discuss the pros and cons of the proposed model. 

4- Whether there is a limit or insufficient method in this paper method?

5- The study lacks a clear comparison between the submitted paper and the more relevant literature contributions, which should highlight the main advantages of the current submission.

6- I find some typos existing in the manuscript. The English needs further improvement.

I find some typos existing in the manuscript. The English needs further improvement.

Reviewer 4 Report

The article investigates the effect of active seat belt pretensioning on driver injuries in unavoidable side crashes in automatic emergency braking (AEB) vehicles.The research work has important implications for vehicle crash safety.However, it is suggested to further modify the article before publication. The specific problems are as follows:

1.1.       It is suggested that the author refer to the literature and reorganize the paper. Cai Z, Xia Y, Bao Z, et al. Creating a human head finite element model using a multi-block approach for predicting skull response and brain pressure[J]. Computer methods in biomechanics and biomedical engineering, 2019, 22(2): 169-179.ï¼› Chang L, Guo Y, Huang X, et al. Experimental study on the protective performance of bulletproof plate and padding materials under ballistic impact[J]. Materials & Design, 2021, 207: 109841.

2.Figure 8 shows adding units.

3.Figure 9 should be redrawn to remove the background colour.

Suggest that the author read through the whole text to make grammatical corrections.
